# Single- and Double-Charge Exchange Reactions and Nuclear Matrix Element for Double-Beta Decay

Hiroyasu Ejiri

Research Center for Nuclear Physics, Osaka University, Osaka 567-0047, Japan; ejiri@rcnp.osaka-u.ac.jp

**Abstract:** Neutrino properties such as the Majorana nature and the masses, which go beyond the standard model, are derived from the experimental double-beta decay (DBD) rate by using the DBD nuclear matrix element (NME). Theoretical evaluations for the NME, however, are very difficult. Single-charge exchange reactions (SCERs) and double-charge exchange reactions (DCERs) are used to study nuclear isospin ($\tau$) and spin ($\sigma$) correlations involved in the DBD NME and to theoretically calculate the DBD NME. Single and double $\tau\sigma$ NMEs for quasi-particle states are studied by SCERs and DCER. They are found to be reduced with respect to the quasi-particle model NMEs due to the $\tau\sigma$ correlations. The impact of the SCER- and DCER-NMEs on the DBD NME is discussed.

**Keywords:** neutrinoless double-beta decay; neutrino mass; nuclear matrix element; isospin spin correlations; single- and double-charge exchange reactions; quenching of the weak coupling $g_A$





## 1. Introduction

Neutrinoless double beta decay (DBD) is one of very powerful and realistic probes for studying neutrino properties such as the neutrino nature (Dirac or Majorana), the neutrino mass scale and the mass hierarchy, the possible right-handed weak interaction, and others, which are beyond the standard electro-weak model. Recent experimental and theoretical studies of DBDs are given in review articles [1–3] and references therein.

The neutrino mass and the other neutrino properties beyond the standard model are derived from the DBD transition rate $T^{0\nu}$ by using the nuclear matrix element (NME). In case of the light neutrino-mass process, the rate is given in terms of the effective neutrino mass $m_\nu$ and the NME $M^{0\nu}$ as $T^{0\nu} = G|m_\nu M^{0\nu}|^2$, where $G$ is the kinematic (phase space) factor, including the axial-vector weak coupling term of $g_A^4$. Thus, one needs an accurate value for the $M^{0\nu}$ of the nuclear physics quantity in order to study the neutrino properties of the particle physics interest from the observed rate $T^{0\nu}$. A theoretical evaluation for the $M^{0\nu}$, however, is extremely difficult since it is very sensitive to the nuclear model to be used for the calculation of nuclear parameters such as the nuclear correlations and the quenching of the axial-vector coupling to be used for the model calculation. There is no way to experimentally measure the value for $M^{0\nu}$. Therefore, it is interesting to experimentally study nuclear correlations and nuclear parameters involved in $M^{0\nu}$ to provide experimental inputs to help theoretical evaluations for $M^{0\nu}$. Experimental and theoretical studies of the DBD NMEs are discussed in review articles [4–8] and references therein. Recent DBD experiments are given in [9], and recent experimental approaches to neutrino responses for single- and double-beta decays are discussed in [10].

The present report aims to show that single- and double-charge exchange reactions (SCERs and DCERs) with mediate energy ($E \approx 0.1$ GeV per nucleon) projectiles are used to study the isospin($\tau$) spin($\sigma$) strengths in the wide excitation and momentum regions, which are associated with the DBD NME, and thus help to theoretically evaluate the DBD NME. Here, note that SCER and DCER at the medium-energy preferentially excite the $\tau\sigma$ modes relevant to single and double $\beta$ decays.

SCER, DCER, and DBD are schematically illustrated in Figure 1. The strong $\tau\sigma$ interaction pushes up the $\tau\sigma$ Gamow–Teller (GT $J^\pi = 1^+$) and spin-dipole (SD $J^\pi = 2^-$) strengths to form the GT and SD giant resonances (GT-GR and SD-GR) in the high-excitation region, leaving little $\tau\sigma$ GT and SD strengths in the ground and low-lying quasi-particle (QP) states in the intermediate nucleus of $^A_{Z+1}$X. Likewise, it pushes up the double $\tau\sigma$ GT and SD strengths to form the GT and SD double giant resonances (GT-DGR and SD-DGR) in the high-excitation region, leaving little double $\tau\sigma$ GT and SD strengths in the ground and low-lying QP states in the final nucleus $^A_{Z+2}$X. Accordingly, the $\tau\sigma$ axial-vector DBD NMEs for the ground and low-lying QP states are significantly reduced.

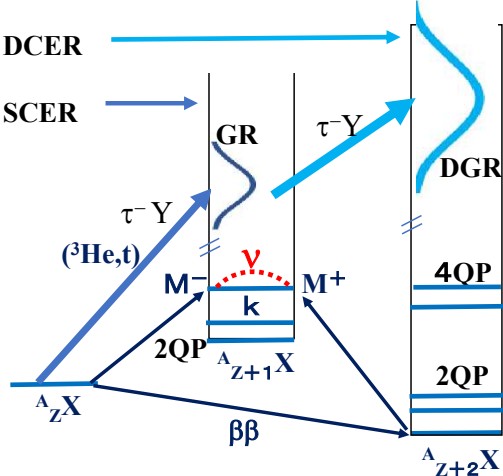

**Figure 1.** Schematic diagram for the $0\nu\beta\beta$ DBD transition of $^A_Z$X$\rightarrow^A_{Z+2}$X with a neutrino exchange. SCER: $^A_Z$X$\rightarrow^A_{Z+1}$X. DCER: $^A_Z$X$\rightarrow^A_{Z+2}$X. QP: quasi-particle state. GR: giant resonance. DGR: double giant resonance. M$^-$ (M$^+$): $\tau^-$ ($\tau^+$) single-$\beta$ response associated with DBD.

So far, the SCERs of ($^3$He,$t$) at RCNP have extensively been used to study GT and SD NMEs in the intermediate nuclei, which are associated with the main components of two-neutrino and neutrinoless DBD NMEs, respectively [8,11,12]. In fact, the GT (s-wave) NMEs in the QP region of 0–6 MeV, which are the main components of the two-neutrino DBD within the standard model, have been well studied previously in a series of the SCERs [8]. On the other hand, the SD (p-wave) NMEs are the main components of the neutrinoless DBD beyond the standard model [8,13]. Recently, SD giant resonances at the high-excitation region have been shown to be closely related to the DBD NMEs [14].

The present work studies for the first time (i) the SD strength at the QP and low-excitation regions by the intermediate-energy SCERs on DBD nuclei, (ii) the GT-SD strength at the QP region via the intermediate-energy DCER, and (iii) the reduction in SCER SD strength due to the $\tau\sigma$ correlation and that of the DCER GT-SD strength due to the double $\tau\sigma$ correlations. Then, we discuss the impact of the present findings on the DBD NMEs.

The $0\nu\beta\beta$ NME is expressed as [4,8]

$$M^{0\nu} = \left(\frac{g_A^{eff}}{g_A}\right)^2 [M_{GT}^{0\nu} + M_T^{0\nu}] + \left(\frac{g_V^{eff}}{g_A}\right)^2 M_F^{0\nu}, \tag{1}$$

where $M_{GT}^{0\nu}$, $M_T^{0\nu}$, and $M_F^{0\nu}$ are the Gamow–Teller (GT), tensor (T), and Fermi (F) NMEs, respectively. Since the axial-vector coupling term of $g_A^4$ is included convensionally in the phase space factor $G$, $M^{0\nu}$ includes in the denominator the coupling term of $g_A^2$. We use here the free-nucleon coupling of $g_A = 1.27$, and introduce the effective axial-vector and vector couplings of $g_A^{eff}$ and $g_V^{eff}$ for nuclear effects, all in units of the vector coupling $g_V$ for a free nucleon. The $M^{0\nu}$ is given by the sum of the NME $M^{0\nu}(i)$ for all the intermediate states (i).

The DBD nuclei to be used for DBD experiments are medium-heavy nuclei of many body-interacting hadron (nucleon, meson, isobar) systems, and the DBD NMEs are very sensitive to nuclear interactions and correlations among them, and thus accurate theoretical evaluations for them are extremely difficult [8]. Then, the effective weak couplings of $g_A^{eff}$ and $g_V^{eff}$ are introduced to incorporate the nuclear and non-nuclear correlations and nuclear medium effects, which are not explicitly included in the model calculations. Note that the largest NME in Equation (1) is the GT NME with the $\sigma\tau$ operators, and the dipole (SD) component with the orbital angular momentum $L = 1$ is one of the main components of $M_{GT}^{0\nu}$ since it matches with the momentum of the virtual neutrino exchanged between two nucleons in the intermediate DBD nucleus (see Figure 1).

## 2. Single Charge-Exchange Reaction

Let us discuss first the ($^3$He,$t$) SCERs for the low-lying (0–2 MeV) SD QP states in DBD nuclei. The medium-energy (0.42 GeV) projectile of $^3$He is provided from the RCNP cyclotron, and the emitted $t$ was momentum-analyzed by the RCNP spectrometer Grand-RAIDEN [8]. The SCER energy spectrum for $^{130}$Te [15] is shown as an example of the $\tau\sigma$ strength distributions in Figure 2. Here, most GT and SD strengths are located in the high-excitation GR regions at $E$(GT-GR) $\approx$ 14 MeV and $E$(SD-GR) $\approx$ 22 MeV, respectively. Little GT and SD strengths are seen at the low-excitation QP region. The F-GR is the Fermi-type GR, called the isobaric analogue state (IAS), due to the $\tau\tau$ interaction. It collects all F strengths, leaving no strength at the low-excitation region.

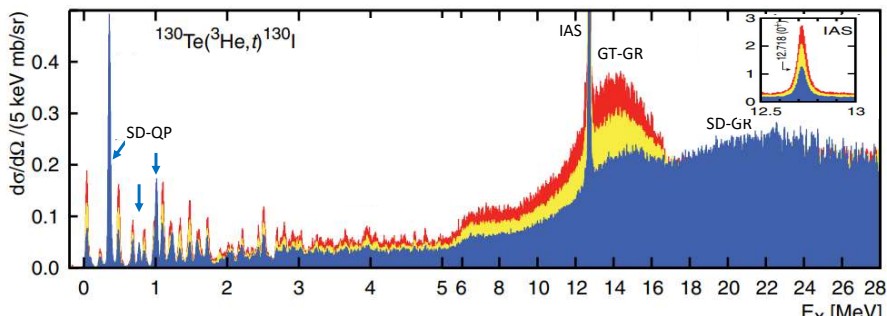

**Figure 2.** The energy spectrum of $^{130}$Te($^3$He,$t$)$^{130}$I [15]. IAS, GT-GR, and SD-GR are Fermi $0^+$, GT $1^+$, and SD $2^-$ giant resonances, respectively. The energy scale below 5 MeV is enlarged by a factor 2 to make the sharp peaks visible. F and GT transitions with $L = 0$ are enhanced at the forward angles of 0–0.5 deg. (red) and 0.5–1 deg. (yellow), while the SD ones with ($L = 1$) at a larger angle of 1–1.5 deg. (blue). Thick blue arrows: SD QP states.

In the present work, we first obtain the SD strengths and the SD NMEs for the low-lying QP states by referring to the well-known IAS F strength, and then compare the experimental SD NMEs with the QP model NMEs to extract the reduction coefficients $k_{\sigma\tau}$(SD) and $k'_{\sigma\tau}$(SD) for them.

The SD cross-section is expressed in terms of the SD strength $B$(SD,QP$_i$) as

$$d\sigma(\text{SD}, \text{QP}_i)/d\Omega = (2L+1)K(\text{SD}, \text{QP}_i)N(\text{SD}, \text{QP}_i)|j_1(q_iR)|^2|J_{\tau\sigma}|^2 B(\text{SD}, \text{QP}_i), \quad (2)$$

where $K(\text{SD}, \text{QP}_i)$ and $N(\text{SD}, \text{QP}_i)$ are the kinematic and distortion factors, respectively; the coefficient $(2L + 1)$ is 3 for the SD ($L = 1$) excitation; $j_1(q_iR)$ is the spherical Bessel function with $q_i$ and $R$ being the linear momentum transfer and the effective nuclear radius, respectively; and $J_{\tau\sigma}$ is the volume integral of the $\tau\sigma$ interaction, respectively. The projectile SCER strength for $^3$He $\to t$ is included in the kinematic factor. The strength is expressed by using the SD NME $M(\text{SD}, \text{QP}_i)$ as $B(\text{SD}, \text{QP}_i) = |M(\text{SD}, \text{QP}_i)|^2$ in the present case of the even–even target nucleus. The transition operator is given as $T_{KLS} = i^L[\sigma_S \times Y_L]_K$ with

$K = 1, 2, L = 1$, and $S = 1$. In fact, the matrix element $<i | T_{211} | f>$ for the spin-stretched transition of $l \pm 1/2 \to (l \pm 1) \mp 1/2$ is dominant.

Here, we use the differential cross-section $d\sigma(\text{F}, \text{IA})/d\Omega$ for IAS as a reference cross-section, and we discuss the differential cross-section ratio

$$\frac{d\sigma(\text{SD}, \text{QP}_i)/d\Omega}{d\sigma(\text{F}, \text{IA})/d\Omega} = 3 \frac{|j_1(q_i R)|^2}{|j_0(q_{\text{IA}} R)|^2} \frac{|J_{\tau\sigma}|^2}{|J_\tau|^2} \frac{B(\text{SD}, \text{QP}_i)}{B(\text{F}, \text{IA})}, \tag{3}$$

where $J_\tau$ is the volume integral of the $\tau$ interaction and the F strength for IAS is $B(\text{F}, \text{IA}) = N - Z$, with $N, Z$ being the neutron and proton numbers. Here, the ratios for the kinematic factors and the distortion factors are assumed to be 1 for the present medium-energy reaction since the energies and the momenta of the incoming and outgoing particles for the QP states are almost same as those for the IAS in the present SCER with the sub-GeV energy and the GeV/c momentum.

The ratio of the $\tau\sigma$ to $\tau$ interaction-integral squares is $|J_{\tau\sigma}|^2/|J_\tau|^2 \approx (V_{\tau\sigma}/V_\tau)^2 \approx 9.8$ [16,17] for the present medium-energy projectile. Then, one obtains the preferential GT and SD excitations by the large $\tau\sigma$ interaction. The differential cross-section for the low-lying SD QP state obtains the maximum value at $\theta \approx 2$ deg., while the one for the IAS at $\theta \approx 0$ deg. So, we use the ratio of the SD to F cross-sections at these $\theta \approx 2$ deg. and 0 deg., respectively.

The SD strength $B(\text{SD}, \text{QP}) = \sum_i B(\text{SD}, \text{QP}_i)$ is obtained from the experimental SD cross-sections for the low-lying ($E_X = 0$–2 MeV) QP states excited by the SCERs on $^{76}$Ge, $^{82}$Se, $^{96}$Zr, $^{100}$Mo, $^{128}$Te, $^{130}$Te, and $^{136}$Xe, which are all typical DBD nuclei of the current interest [15,18–22]. The obtained SD and F strengths are shown in Table 1.

Let us compare the obtained SD strength with the simple QP model strength $B_{QP}(\text{SD}, \text{QP}) = \sum_i B_{QP}(\text{SD}, \text{QP}_i)$, with $B_{QP}(\text{SD}, \text{QP}_i) = |M_{QP}(\text{SD}, \text{QP}_i)|^2$ being the square of the QP model NME for the $\text{QP}_i$ state. We introduce a reduction coefficient defined as $k_{\tau\sigma}(\text{SD}) = [B(\text{SD}, \text{QP})/B_{QP}(\text{SD}, \text{QP})]^{1/2}$. The obtained coefficients for DBD nuclei are shown in the fourth column of the Table 1, and are plotted in Figure 3. The QP SD NMEs are effectively reduced by the reduction coefficient $k_{\tau\sigma}(\text{SD}) \approx 0.3$ due to the $\tau\sigma$ correlations and others with respect to the QP model SD NMEs without the $\tau\sigma$ and other correlations. Similar reductions are seen in single-$\beta$ SD NMEs for QP SD states in medium-heavy nuclei [13].

**Table 1.** Top: The SD strengths and the reduction coefficients for the QP and low-lying states by SCERs on DBD nuclei. The SD strength $B(\text{SD,QP})$, the F strength $B(\text{F,IA})$, the reduction coefficient $k_{\tau\sigma}(\text{SD})$ are shown for the QP states, and the reduction coefficient $k'_{\tau\sigma}(\text{SD})$ for the low-lying states. See text. Bottom: the GT-SD strength and the reduction coefficient for the QP states by DCER on $^{56}$Fe. The GT-SD strength $B(\text{GTSD,QP})$, the F strength $B(\text{FF,DIA})$, and the reduction coefficient $k_{\tau\sigma}(\text{GTSD})$ are shown.

| Nuclide | $B(\text{SD, QP})$ | $B(\text{F, IA})$ | $k_{\tau\sigma}(\text{SD})$ | $k'_{\tau\sigma}(\text{SD})$ |
|---|---|---|---|---|
| $^{76}$Ge | $0.080 \pm 0.016$ | 12 | $0.30 \pm 0.05$ | $0.26 \pm 0.05$ |
| $^{82}$Se | $0.091 \pm 0.018$ | 14 | $0.29 \pm 0.04$ | - |
| $^{96}$Zr | $0.024 \pm 0.005$ | 16 | $0.27 \pm 0.04$ | $0.31 \pm 0.06$ |
| $^{100}$Mo | $0.053 \pm 0.011$ | 16 | $0.35 \pm 0.05$ | $0.33 \pm 0.06$ |
| $^{128}$Te | $0.452 \pm 0.090$ | 24 | $0.32 \pm 0.05$ | $0.29 \pm 0.05$ |
| $^{130}$Te | $0.456 \pm 0.090$ | 26 | $0.31 \pm 0.05$ | $0.29 \pm 0.05$ |
| $^{136}$Xe | $0.457 \pm 0.091$ | 28 | $0.34 \pm 0.05$ | $0.26 \pm 0.05$ |
| Nuclide | $B(\text{GTSD,QP})$ | $B(\text{FF, DIA})$ | $k_{\tau\sigma}(\text{GTSD})$ | - |
| $^{56}$Fe | $0.61 \pm 0.12$ | 8 | $0.092 \pm 0.014$ | - |

Now we compare the summed cross-section $\sigma_L(\text{SD})$ for low-lying SD ($L = 1$) states in the excitation region of 0–12 MeV below the SD giant resonance with the total cross-section $\sigma_T(\text{SD})$, including the giant resonance. In fact, the SD states are not well separated in the 4–12 MeV region, and thus the SD component is derived from the multi-

pole de-composition of the measured angular distribution. Then, the ratio is given as $d\sigma_L(\text{SD})/d\sigma_T(\text{SD}) = (k'_{\sigma\tau}(\text{SD}))^2$ with $k'_{\tau\sigma}(\text{SD})$ being the effective reduction coefficient for the low-lying $L = 1$ states. The obtained values for $k'_{\sigma\tau}(\text{SD})$ are shown in the fifth column of Table 1 and in Figure 3. Note that $k'_{\sigma\tau}(\text{SD}) \approx 0.3$ is nearly the same as $k_{\sigma\tau}(\text{SD})$.

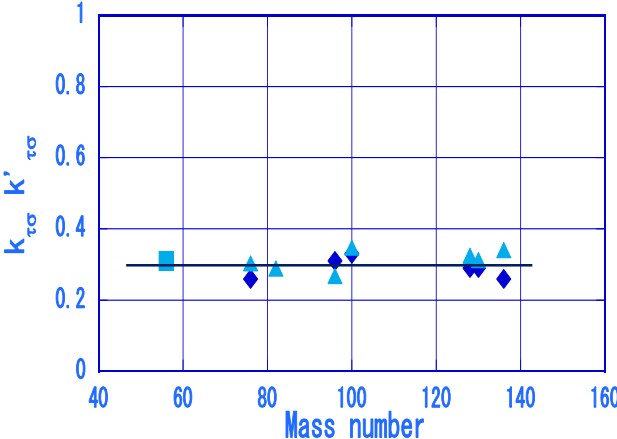

**Figure 3.** Reduction coefficients for axial-vector NMEs. Light blue triangles: $k_{\tau\sigma}(\text{SD})$ for the QP SD states by SCERs on DBD nuclei. Blue diamonds: $k'_{\tau\sigma}(\text{SD})$ for low-ling SD states by SCERs on DBD nuclei. Light blue square: $(k_{\tau\sigma}(\text{GTSD}))^{1/2}$ for the QP GT-SD states by DCER on $^{56}$Fe. Solid line: the reduction coefficient of 0.3 to guide eye.

## 3. Double Charge-Exchange Reaction

Next, we discuss the GT-SD strengths from the DCERs. They are derived for the first time from the GT-SD cross-sections and are compared with the QP model GT-SD NMEs to extract the reduction coefficient $k_{\sigma\tau}(\text{GTSD})$ due to the double $\sigma\tau$ correlations. DCERs using heavy ions are interesting for studying DBD NMEs [23–26]. The NEUMEN project is an extensive DCER program for NMEs [27]. Double GT NMEs are related with the DBD NMEs [28].

The DCER $^{56}$Fe($^{11}$B,$^{11}$Li)$^{56}$Ni is measured at the forward angles of $\theta = 0$–2.5 deg. by using the medium-energy (0.88 GeV) $^{11}$B projectile [24]. The accelerator and the spectrometer used for the DCER are the same as for the SCER at RCNP. The reaction particle of $^{11}$Li is identified by measuring the TOF (time of flight) and the energy loss by the plastic scintillators at the focal plane of the spectrometer. Merits of this reaction are (a): $^{11}$B is the lightest projectile to be used for DCERs, (b): the $\tau\sigma$ interaction is dominant at this medium energy to preferentially excite axial-vector GT and SD states, and (c): the ground state of $^{11}$Li is only the bound state and thus the energy spectrum of the $^{11}$Li reflects the excitation spectrum for the residual nucleus of $^{56}$Ni.

The observed energy spectrum of the DCER on $^{56}$Fe is shown in Figure 4, together with the one on $^{13}$C and the SCER on $^{76}$Ge at $\theta = 2$–2.5 deg., for comparison. The double IAS is clearly observed in the DCER on $^{56}$Fe at $E(\text{DIA}) = 15$ MeV. The DCER strength for the ground and low-lying QP states below $E \approx 15$ MeV are much smaller than the strength for DIAS, as shown in Figure 4C. This feature is just the same as in the SCER (see Figure 4A). On the other hand, the ground and low-lying states are well excited in the DCER on $^{13}$C, as shown in Figure 4B.

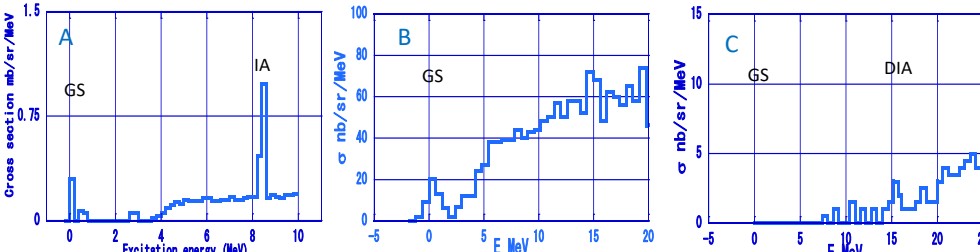

**Figure 4.** Energy spectra of the SCER and the DCERs. (**A**): SCER of ($^3$He,$t$) on $^{76}$Ge at $\theta$ = 2–2.5 deg., where the SD ($L$ = 1) excitation obtains maximum [18]. (**B**): DCER of ($^{11}$B,$^{11}$Li) on $^{13}$C at $\theta$ = 0–2.5 deg., where GTSQ ($L$ = 2) excitation obtains maximum [24]. (**C**): DCER of ($^{11}$B,$^{11}$Li) on $^{56}$Fe at $\theta$ = 0–2.5 deg., where GTSD ($L$ = 1) excitation obtains maximum [24]. GS: Ground state. IA: isobaric analogue state. DIA: double isobaric analogue state.

DCER on $^{56}$Fe for the QP states below 15 MeV is considered to proceed by two steps of the GT and SD excitations, and vise versa. It is noted here that the reaction $Q$-value is around $Q \approx -50$ MeV for the residual states around 10 MeV, and thus the angular and linear momentum transfers are around $L \approx 1$ and $q \approx 120$ MeV/c. Thus, the GT and SD double-charge exchange reactions with the total $L = 1$ transfer is favored, but the double IAS, the double GT, and the ground state ($J^\pi = 0^+$) DCERs with the total $L = 0$ transfer are dis-favored. The DCER differential cross-section for the $k$th QP state is expressed in terms of the DCER strength $B(\text{GTSD},\text{QP}_k)$ as

$$d\sigma(\text{GTSD},\text{QP}_k)/d\Omega = (2L + 1)K(\text{GTSD},\text{QP}_k)N(\text{GTSD},\text{QP}_k)|j_1(q_kR)|^2|J_{\tau\sigma}|^4 B(\text{GTSD},\text{QP}_k), \quad (4)$$

where $K(\text{GTSD},\text{QP}_k)$ and $N(\text{GTSD},\text{QP}_k)$ are the kinematic and distortion factors, $2L + 1 = 3$ for the GTSD ($L = 1$) excitations, $j_1(q_kR)$ is the spherical Bessel function with $q_k$ and $R$ being the linear momentum transfer and the effective nuclear radius for DCER, and $J_{\tau\sigma}$ is the volume integral of the $\tau\sigma$ interaction. Note here that the projectile DCER strength for $^{11}$Be$\rightarrow^{11}$Li is included in the kinematic factor. The strength $B(\text{GTSD, QP}_k)$ is given by the square of the NME $|M(\text{GTSD},\text{QP}_k)|$. The NME is expressed as $M = \sum_J <|\text{T}_{101}\text{T}_{J11}|>$ with the GT transition operator $\text{T}_{101} = \tau\sigma$ and the SD one $\text{T}_{J11} = \tau[\sigma\times\text{Y}_1]_J$.

Here, we use the differential cross-section $d\sigma(\text{FF,DIA})/d\Omega$ for the double IAS (DIA) with the well-known strength $B(\text{FF, DIA}) = 4 \times 2$ as a reference cross-section, as used in SCER. The differential cross-section ratio is

$$\frac{d\sigma(\text{GTSD},\text{QP}_k)/d\Omega}{d\sigma(\text{FF},\text{DIA})/d\Omega} = \frac{3|j_1(q_kR)|^2}{|j_0(q_{DI}R)|^2} \frac{|J_{\tau\sigma}|^4}{|J_\tau|^4} \frac{B(\text{GTSD},\text{QP}_k)}{B(\text{FF},\text{DIA})}, \quad (5)$$

where the ratios for the kinematic factors and the distortion factors are assumed to be 1 for the present medium-energy (0.88 GeV) DCER, as in case of the medium-energy SCERs.

The ratio of the interaction integrals is given as $|J_{\tau\sigma}|^4/|J_\tau|^4 \approx (V_{\tau\sigma}/V_\tau)^4 \approx 11.5$ [16,17]. Thus, the axial-vector ($\tau\sigma$) excitation is dominant in the present DCER with the medium-energy projectile as in the SCERs. Since the QP states $\text{QP}_k$ in the final nucleus of $^{56}$Ni are not separated from each other, the summed cross-section of $d\sigma(\text{GTSD},\text{QP})/d\Omega = \sum_k d\sigma(\text{GTSD},\text{QP}_k)/d\Omega$ up to 15 MeV is used to obtain the summed strength of $B(\text{GTSD},\text{QP}) = \sum_k B(\text{GTSD},\text{QP}_k)$. The obtained strength of $B(\text{GTSD, QP}) \approx 0.061$ is two orders of the magnitude smaller than the summed QP-model strength of $B_{QP}(\text{GTSD, QP}) = 6.6$. Noting that the GTSD NMR for the DCER is given as $M(\text{GTSD},\text{QP}_k) = (B(\text{GTSD},\text{QP})_k)^{1/2}$, the DCER NMEs for the QP states are reduced effectively by the coefficient of $k_{\tau\sigma}(\text{GTSD}) = (0.061/6.6)^{1/2} \approx 0.1$. This is close to the square of $k_{\tau\sigma}(\text{SD}) \approx 0.3$ for the SCER. The DCER NMEs for QP states are reduced doubly by the coefficient of $k_{\tau\sigma}(\text{SD})^2 \approx 0.1$ due to the double $\tau\sigma$ correlations and others in medium-heavy nuclei. The ground state of $^{13}$O is well excited via the $\text{p}_{1/2}(\text{n})\rightarrow\text{p}_{1/2}(\text{p})$ GT and the $\text{p}_{3/2}(\text{n})\rightarrow\text{p}_{1/2}(\text{p})$ SQ (spin quadrupole) excitations in case of the DCER on the light nucleus of $^{13}$C (Figure 4B).

## 4. Concluding Remarks

In short, the SD NMEs and the GT-SD NMEs for the low-lying QP states, which are derived from the SCERs and DCER, are found to be reduced with respect to the QP model NMEs by $k_{\sigma\tau}(\text{SD}) \approx 0.3$ and $k_{\sigma\tau}(\text{GTSD}) \approx 0.1$, respectively. This indicates the similar severe reductions in the single-beta and double-beta strengths for the low-lying QP states. The reduction is partly due to the repulsive nuclear $\tau\sigma$ correlation that is well included in pnQRPA and partly to the non-nucleonic (mesons and isobars) and nuclear medium effects that are not explicitly included in pnQRPA. The effects which are not in the pnQRPA are incorporated by introducing the quenched axial-weak coupling of $g_A^{eff}$ [8,29–33].

It is very interesting to extend the present DCER to DBD nuclei to see how axial-vector DCER strengths are concentrated in the double giant resonance regions around $E \approx 30$–$50$ MeV, resulting in a severe reduction in the DCER NMEs for the ground and low-lying QP states. Then, the DECR data, together with the SCER data, are used to check the theoretical models for DBD NMEs. The single-$\beta$ and $2\nu$-DBD NMES were discussed in [34]. Experimental SCER GT NMEs, which are significantly reduced with respect to the OP model NMEs, are shown to effectively reproduce the observed $2\nu$-DBD NMEs, which are significantly reduced with respect to the QP model NMEs [35].

SCERs and DCERs discussed so far are CERs of light and heavy ions via a strong (nuclear) interaction. CERs with lepton probes of neutrinos and muons via a weak interaction are also used to study neutrino nuclear responses associated with DBDs. The CER of $^A_Z X + \nu \to ^A_{Z+1} I + e$, by using intense $\nu$ projectiles, is used to study the $M^-$ response associated with DBD [36]. Ordinary muon capture (OMC) reaction is a kind of the lepton CER of $^A_{Z+2} Y + \mu^- \to ^A_{Z+1} I + \nu_\mu$, and is used to study the $M^+$ response associated with the DBD NME [37–39]. Similar information is also obtained by using photon probes [8,10]. Experimental studies on these CERs are encouraged.

**Funding:** This research received no external funding.

**Institutional Review Board Statement:** Not applicable.

**Informed Consent Statement:** Not applicable.

**Data Availability Statement:** The data used for the present analyses and discussions are in refs. cited in the text.

**Conflicts of Interest:** The author declares no conflict of interest.

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
