# Peer review of "Single- and Double-Charge Exchange Reactions and Nuclear Matrix Element for Double-Beta Decay"

_universe, doi:10.3390/universe8090457_

Round 1
Reviewer 1 Report
The present manuscript describes the calculation of nuclear matrix elements in the context of double beta decays. The author discusses single and double charge-exchange reactions. The content of this manuscript appears to be interesting and relevant in the context of neutrinoless double beta decay searches.
However, I think the manuscript requires major revision before publication. It is not clear to me, over an extended fraction of the text, to what degree the presented results are a review of previous publications and to what degree the results are original. I recommend to better indicate what additional findings are presented in this text. At the same time, I am missing details on the methodology used in this work. I recommend to elaborate more on how the results were derived.
Apart from these major concerns, I have one small item regarding equation 1. Here, the denominators are not explained well. From the text below, it appears to me as if the whole fraction the effective coupling, but then g_a in the equation versus g_v in the text does not make sense. Please be more clear in this explanation.
Author Response
Thank you for the very valuable comments. The replies for the comments are in a separate file.

Reviewer 2 Report
In this paper the author studied single and double spin-isospin nuclear matrix element for quasi-particle states by single charge-exchange and double charge-exchange reactions. They are reduced uniformly with respect to the quasi-particle model nuclear matrix elements due to the spin-isospin correlations. Also, impact of the single charge-exchange and double charge-exchange reactions nuclear matrix elements on the double beta decay nuclear matrix element is discussed.
I think this paper is interesting and valuable contribution to the field. The article is well written and articulated. The paper can be accepted for publication in Universe without any further changes required from the authors.
Just several small technical misprints:
Page 2, Paragraph 4: "The DBD nuclei to be used for DBD experiments .... nonpnuclear ...." --> "... non-nuclear ..."
Page 3 after Eq. (2): "... for the SD (L-1) excitation ..." --> "... for the SD (L=1) excitation ..."
Page 4, Paragraph 2: "... \sum_i |B_{QP}(SD,QP_i) ... |MQP(SD,QP_i|^2 ..." --> "... \sum_i B_{QP}(SD,QP_i) ... |M_{QP}(SD,QP_i)|^2 ..."
Page 5, Paragraph 1: "... The obtained values for k′(\sigma\tau)) are ..." --> "... The obtained values for k′(\sigma\tau) are..."
Page 6, Paragraph 5: " ... of ligh and heavy ions ..." --> " ... of light and heavy ions ..."
Author Response
Thank you for the comments. The replies are in the attached file.

Round 2
Reviewer 1 Report
I thank the author for the new version of the manuscript which addresses all my concern adequately. I believe the manuscript is ready for publication in its present form. Congratulations.